# A Narrative Review of Continuing Professional Development Needs for Pharmacists with Respect to Pre-exposure Prophylaxis (PrEP) for Human Immunodeficiency Virus (HIV)

**DOI:** 10.3390/pharmacy8020084

**Published:** 2020-05-11

**Authors:** Kyle J. Wilby, Alesha J. Smith

**Affiliations:** School of Pharmacy, University of Otago, Dunedin 9054, New Zealand; alesha.smith@otago.ac.nz

**Keywords:** pharmacy, continuing education, continuing professional development, HIV, pre-exposure prophylaxis, antiretrovirals, education

## Abstract

Pre-exposure prophylaxis (PrEP) for the human immunodeficiency virus (HIV) is rapidly increasing in use worldwide, with many countries now publicly funding use for high risk populations. Pharmacists, as front-line care providers, must have the necessary knowledge, skills and attitudes to effectively provide care to PrEP patients. The aim of this review was to identify priority areas and key gaps for continuing professional development (CPD) needs relating to PrEP for practicing pharmacists. An electronic search of PubMed, EMBASE, International Pharmaceutical Abstracts and CPD-related journals was supplemented with a manual search of references to identify articles describing pharmacists’ knowledge, perceptions and experience with PrEP. A total of eight articles were identified across four countries. Pharmacists were consistently found to lack knowledge and awareness of PrEP, express low confidence/comfort with patient care practices, report a lack of experience and/or intentions to provide patient care, but overall had positive perceptions of PrEP therapy. Older pharmacists with more experience commonly reported greater knowledge gaps than recently trained pharmacists. CPD should therefore aim to increase pharmacists’ baseline knowledge and awareness of PrEP and treatment guidelines, as well as be directed towards older pharmacists with more experience.

## 1. Introduction

Pre-exposure prophylaxis (PrEP) for human immunodeficiency virus (HIV) has become a central component of sexual health care in many countries. The combination of tenofovir and emtricitabine was first approved in the United States (US) by the U.S. Food and Drug Administration (FDA) in 2012 [1]. Since then, numerous countries have approved the use of PrEP for HIV prevention and in 2018, New Zealand was one of the first countries to publicly fund its use [2]. The Centers for Disease Control and Prevention (CDC) suggests PrEP should be considered for HIV-negative people who have had anal or vaginal sex in the past 6 months and have an HIV-positive partner, have not consistency used a condom, or have been diagnosed with a sexually transmitted infection (STI) in the previous 6 months [3]. PrEP may also be considered for people who inject drugs or other high-risk populations. Efficacy rates for preventing transmission are noted to be high for sexual transmission and moderate for transmission through drug use [4,5]. As of April 2020, 75 countries had some form of PrEP registered for use, and 44 of these countries offered funding for high risk populations [6]. It is estimated that over 500,000 patients are current PrEP enrollees; it is therefore essential that pharmacists have the necessary knowledge, awareness, confidence and perceptions to provide optimal care to patients seeking its use.

Continuing professional development (CPD) is used by pharmacists globally to maintain up-to-date knowledge and skills throughout their careers alongside fulfilling mandatory requirements, such re-licensure and/or registration [7]. CPD can take many forms including:online webinars;self-study;conference sessions;workplace-based projects;structured courses.

Effective CPD encourages reflective practice and application of learning which ultimately enhances professional competences and leads to better delivery of health services and patient outcomes relative to the population served [8]. Each pharmacist therefore should essentially have their own individual learning plan which may include; knowledge of new medications or indications, updated treatment guidelines or recommendations for evidence-based practice, skills such as communicating with special populations, knowledge or skills related to leadership or management, population health considerations or policy/governance practices.

Determining the needs of target learners is particularly relevant to PrEP education given the differing rates of uptake and funding across jurisdictions over the last decade. It is likely that pharmacists have varying degrees of experience and knowledge. Helping learners identify their gaps in knowledge and optimizing learning outcomes of any CPD program or activity to meet this need is the key for effective implementation [9].

It is currently unknown what content PrEP-related CPD activities should address when designed for educating the workforce. As more countries publicly fund PrEP for high risk populations (e.g., men who have sex with men, intravenous drug users, sex workers), it is likely that dispensing activity of PrEP will increase further and pharmacists will need to be able to adequately assess patients, manage patients’ therapeutic needs, perform monitoring and follow-up, and provide PrEP-related medicines information to patients and other healthcare professionals [10]. In order to do this, pharmacists must be knowledgeable of PrEP and its components, be aware of treatment guidelines and patient resources available be conscious of the social and health consequences for patients choosing to initiate PrEP. Due to the recent release of PrEP in the last decade, it is likely that most of the pharmacist workforce received little to no training with respect to its use. Although CPD needs have not specifically been addressed to date, a number of studies have been published that report pharmacists’ knowledge, awareness, experience and perceptions of PrEP and its use.

The aim of this review is, therefore, to identify priority areas and key gaps for PrEP-related CPD targeted for practicing pharmacists.

## 2. Data Sources and Methods

A literature search of the electronic databases PubMed, EMBASE and International Pharmaceutical Abstracts (IPA) was conducted from conception until 9 April 2020. Combinations of keywords included: PrEP OR pre-exposure prophylaxis [title/abstract] AND pharmacy OR pharmacist [title/abstract] AND education OR continuing professional development. Three pharmacy education journals (American Journal of Pharmaceutical Education, Currents in Pharmacy Teaching and Learning, Pharmacy Education) were also searched using the keywords PrEP and pre-exposure prophylaxis. References of relevant articles were screened to capture any article not identified in the electronic search.

Articles were included in the review if they reported on CPD activities designed for pharmacists about PrEP, or reported on pharmacists’ knowledge, awareness, confidence/comfortability, experiences or attitudes/perceptions regarding PrEP use. Articles that included pharmacists and other healthcare professionals were included if results could be extracted for pharmacists only. Articles solely including undergraduate pharmacy students or intern pharmacists (pre-licensure/registration) were excluded.

One investigator (KW) extracted data from each study. Data included authors and year of publication, population, setting, aim, study design, outcomes and results. Notes pertaining to study quality were also documented, but studies were not systematically assessed for risks of bias. Extraction was validated by the second investigator (AS). Study outcomes and results were then categorized according to knowledge/awareness, comfort/confidence, intentions/experience and perceptions/attitudes. Any other outcomes and results were categorized as ‘other’. Study results were then synthesized under each category to identify CPD needs for each.

## 3. Results

A total of 8 articles were identified and included in the review. A flowchart of the study selection process is provided in Figure 1. Characteristics of each study are provided in Table 1.

### 3.1. Knowledge/Awareness

Gaps in both knowledge and awareness of PrEP and guidelines published to aid clinical decision making were found within all studies. Sanchez-Rubio Ferrandez et al. found that 46% of hospital pharmacists in Spain had low or no familiarity with PrEP [11]. This compared with only 17% of physicians. Broekhuis et al. found that 42% of midwest pharmacists in the United States were familiar with the use of PrEP but only 25% were familiar with the CDC guidelines for its use [12]. This study also found that older pharmacists were less familiar with PrEP. Meyerson et al. found similar results, with 56% of managing pharmacists from Indiana reporting familiarity with PrEP and 52% of pharmacists reporting they did not have sufficient knowledge (adherence, behavior modification, adverse effects or clinical studies) on the topic [13]. Okoro et al. found 52% of community pharmacists in Minnesota were aware of FDA approval for PrEP with most these respondents reporting to be at least somewhat knowledgeable (in terms of awareness) about PrEP [14]. Only 21% of all respondents reported they had sufficient knowledge to counsel patients on a PrEP prescription. Similar to Broekhuis et al., this study also found older participants to report less knowledge than younger ones [12,14]. Yoong et al. studied PrEP awareness in Canadian HIV pharmacists and found that 94% of those surveyed were somewhat familiar or very familiar with the concept of PrEP [15]. Shaeer et al. found Florida pharmacists to have limited knowledge regarding PrEP [16]. A total of 63% were unaware of CDC guidance on PrEP use and 59% were not aware of what the acronym PrEP represented. Only 29% reported they had sufficient knowledge to counsel patients on PrEP prescriptions (including adherence, behavioral modifications and adverse effects). Of those reporting low levels of knowledge, deficits in knowledge were reported to be across all aspects of PrEP.

Unni et al. completed a knowledge assessment for community pharmacists in Utah [17]. In terms of indication, greater than 75% knew PrEP is indicated for those who have HIV-positive partners yet 55–60% indicated it should be used for those engaging in sex with partners of unknown HIV status, inconsistent condom use, sex workers and injection users. A total of 10% of respondents indicated PrEP should be used for HIV infected individuals themselves, indicating a distinct lack of knowledge. Approximately 46% answered correctly regarding the specific medicines used for PrEP and also for the routine testing required for patients considering PrEP. Despite the deficiencies in knowledge identified, actual knowledge scores were higher than how participants rated their current perceived level of knowledge. Those with a PharmD had higher knowledge scores than those with a bachelor’s degree [17]. Matyanga et al. also completed a knowledge assessment for Zimbabwean pharmacists [18]. Using a scoring system, it was determined that 58% were knowledgeable about PrEP. No association was found with age, gender or years of experience.

### 3.2. Comfort/Confidence

Six studies reported outcomes related to pharmacists’ comfort and/or confidence in performing professional activities relating to PrEP. Broekhuis et al. found >80% of Midwest pharmacists were comfortable assessing HIV risk, performing and counseling on HIV testing and providing counseling on PrEP use [12]. If necessary, however, 30% were not comfortable performing urine-based pregnancy and/or STI testing. Meyerson et al. found 54% of managing pharmacists from Indiana were comfortable counseling PrEP patients [13]. Furthermore, 86% were comfortable dispensing to anyone with a medical need and prescription. Fewer years of practice and having a PharmD were associated with high comfort in dispensing. Higher comfort in counseling was associated with awareness of PrEP, continuing education for PrEP and HIV management and previous experience with PrEP. Okoro et al. found having some knowledge of PrEP was associated with being comfortable counseling PrEP for community pharmacists in Minnesota [14]. Pharmacists were least comfortable counseling about the medications (32%) and behavior modification strategies (20%). Only 1% believed counseling about PrEP was not a role for pharmacists. Yoong et al. reported a total of 31% of HIV pharmacists in Canada to have insufficient familiarity to provide counselling [15]. Unni et al. found approximately 20% of community pharmacists in Utah agreed they had the ability to counsel and had the confidence to counsel [17]. Shaeer et al. found 53% of Florida pharmacists sampled were comfortable counselling patients on PrEP [16]. Pharmacists were most uncomfortable counselling on behavior modifications (31%) and clinical trial data (22%).

### 3.3. Intentions/Experience

Seven studies reported outcomes relating to pharmacists/intentions and/or experiences for provision of PrEP therapy. Broekhuis et al. reported 12% of Midwest pharmacists had previously counseled on PrEP [12]. A total of 54% indicated they were likely to provide PrEP services. Those more likely to indicate provision of services included those who had prior experience counseling HIV patients or previous PrEP counseling, or if they had received previous PrEP continuing education. Meyerson et al. found 16% of managing pharmacists from Indiana reported dispensing PrEP at their pharmacy and 12% had consulted about it [13]. PrEP dispensing was associated with prior continuing education. Okoro et al. found that 72% of community pharmacists in Minnesota reported never receiving inquiries about PrEP from clients and 81% reported never receiving inquiries from providers [14]. A total of 33% of respondents had ever dispensed PrEP. Yoong et al. reported a total of 42% of HIV pharmacists in Canada had received a question about PrEP in the past year [15]. With respect to positively providing education about the use of PrEP, 69% agreed, 4% would not and 27% were uncertain. Unni et al. reported only 17% of community pharmacists in Utah agreed or strongly agreed when asked about the intention to counsel and a further 47% disagreeing [17]. After asking about their confidence to counsel, however, 45% agreed to the final question of ‘I plan to counsel’. Those with <10 years’ experience had greater intentions to counsel than those with more experience. Shaeer et al. reported >80% of Florida pharmacists having no inquiries from either patients or providers about PrEP [16]. Only 22% of those practicing in a community or outpatient setting had dispensed PrEP. Matyanga et al. determined that 94% of pharmacists were willing to stock PrEP and 87% intended to dispense PrEP if provided as a pharmacist-initiated medicine [18].

### 3.4. Perceptions/Attitudes

Seven studies reported outcomes relating to perceptions and/or attitudes. Sanchez-Rubio Ferrandez et al. found that hospital pharmacists in Spain required PrEP to have a minimum efficacy percentage of 86% before supporting its widespread use [11]. Prior to education regarding clinical data, 40% of pharmacists supported its use but this increased to 50% after education was provided. A total of 85% of pharmacists disagreed that PrEP should be publicly reimbursed. Finally, barriers to use were perceived to be increased high-risk behaviors, increases in STIs, drug resistance and cost. Broekhuis et al. reported Midwest pharmacists were mostly concerned about PrEP in terms of time burden (61%), compensation (55%), skill set (39%), patient adherence (63%), loss to follow-up (56%) and promotion of drug resistance (51%) [12]. Only 13% of pharmacists expressed ethical concerns. Okoro et al. found that 56% of community pharmacists in Minnesota disagreed that PrEP use would contribute to increased HIV transmission [14]. Unclear responses were given regarding the use of PrEP and contribution to increased rates of other STIs (47% indicated neither agree nor disagree). A total of 3% indicated they were not willing to counsel on PrEP. Yoong et al. found HIV pharmacists in Canada required PrEP to have a minimal efficacy percentage of at least 70% to recommend [15]. A total of 15% were in favor of reimbursement with no restrictions and 71% agreed public reimbursement could occur if the patient belonged to a high-risk population. Only 25% agreed that all physicians should be able to provide PrEP, with higher rates (>70%) obtained for infectious disease physicians or those practitioners with HIV or STI care familiarity, including nurse practitioners and pharmacists. Unni et al. reported that >90% of community pharmacists in Utah agreed that counseling about PrEP is an ethical obligation and patients would benefit from counseling [17]. Shaeer et al. found 68% of Florida pharmacists perceived that PrEP encouraged high-risk behavior, 65% agreed PrEP would increase rates of other STIs and 26% reported PrEP would lead to increased rates of HIV [16]. Matyanga et al. found most (77%) of pharmacists disagreed that PrEP would lead to long term side effects [18]. A total of 73% believed patients would abandon safe sex practices if taking PrEP and 62% agreed that PrEP would empower women.

### 3.5. Other

Three studies reported preferences for continuing education material. Okoro et al. reported that community pharmacists in Minnesota were seeking data regarding the appropriate high-risk populations for which PrEP is intended [14]. The majority (63.4%) indicated online continuing education as their preferred method of training on PrEP. Yoong et al. found that HIV pharmacists in Canada preferred to receive information via continuing education events or workshops (73%), online learning modules (67%) and/or access to papers (63%) [15]. Shaeer et al. reported Florida pharmacists prefer live continuing education (33%), online continuing education (22%) and CDC guidelines or FDA updates (21%) for CPD [16].

## 4. Discussion

PrEP is becoming a mainstay therapy for prevention of HIV transmission worldwide and many countries are beginning to publicly reimburse its use for high risk populations [6]. As such, pharmacists must be prepared to dispense PrEP, offer counseling to PrEP patients and provide medicines information to patients and other healthcare providers. The aim of this review was to identify priority areas for CPD for practicing pharmacists by identifying studies that assessed pharmacists’ knowledge, confidence, experience and perceptions of PrEP use. Findings from the studies identified demonstrated gaps in knowledge about PrEP, links between a pharmacist’s perceived knowledge/awareness of PrEP and their comfort or confidence in dispensing or counseling, mixed intentions and experience regarding provision of PrEP services and generally positive perceptions/beliefs regarding its use.

A key finding from this review was that pharmacists’ knowledge and awareness of PrEP appeared to influence their confidence in provision of services and intentions for providing these services. CPD programming could therefore be targeted to increase knowledge and awareness of PrEP, including target areas of medicines information, specific indications, guidelines for use and appropriate high-risk populations. While some of these target areas (e.g., medicines information, high risk populations) may be applicable to national or international audiences, others (e.g., indications, guidelines) may be country, state or region-specific. CPD programming may therefore need to be flexible and tailored appropriately, depending on the target of information provided.

A common theme identified from the studies included in the review was that younger pharmacists and those with a PharmD degree had more knowledge and confidence and greater intentions to provide PrEP related services [12,13,14,17]. As the PharmD degree is the current entry-to-practice degree for the United States, it is likely that these two findings are inter-related, with PharmD holders being younger in general. CPD programming should therefore be targeted towards older pharmacists, who may not have had previous training in HIV or sexual health.

Effective CPD should also be designed based on how these pharmacists prefer to learn, which may not always be the same for everyone. Few details were provided about how pharmacists had gained their knowledge nor discussion on effective means to do so. It is clear that basic knowledge is lacking for pharmacists, flexible CPD courses that fill this gap, but still allow clinicians to adapt the information and apply it to their setting is needed [19]. The literature has identified technology-based approaches such as mhealth and ehealth, may be useful to help increase PrEP awareness among patients, this could also be easily adopted for health professionals [20]. Research identifying the best methods for maximum knowledge gain and impact would be of value, re-auditing knowledge at regular intervals would also be useful to determine if more experience (as the number of patients increase) has led to further knowledge and to understand the best CPD tools. Identifying PrEP use through prescribing claims may also help implementation efforts and direct education messages for health professionals that is specific for each region.

The results of this review should be interpreted in light of some limitations. Only eight studies were identified and most (n = 5) were from the United States. This many reduce generalizability of findings to other countries and settings. A second limitation was that the studies were published at different times with respect to PrEP approval and use. Some studies were conducted before PrEP was approved for use in their country and others were conducted afterwards. This may influence outcomes relating to knowledge, confidence and intentions to provide PrEP-related services. Any CPD program should therefore supplement findings from this review with a needs assessment of the population being targeted for education.

## 5. Conclusions

As the number of users of PrEP worldwide surpasses half a million, urgent attention is needed to improve pharmacists’ knowledge and skills to ensure the best clinical practice in the provision, monitoring and support of pre-exposure prophylaxis (PrEP) for the prevention of HIV acquisition. CPD should aim to increase pharmacists’ baseline knowledge and awareness of PrEP and treatment guidelines, which should improve confidence and intentions to provide services. CPD should also be targeted towards older pharmacists with more experience. Future studies should reassess pharmacists’ knowledge, confidence, experience and perceptions with PrEP, especially as its use increases worldwide.

## Figures and Tables

**Figure 1 pharmacy-08-00084-f001:**
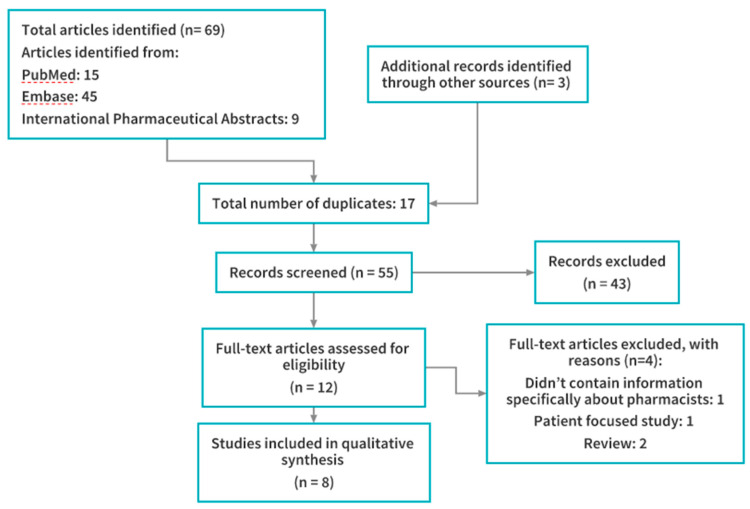
Flowchart of study selection and inclusion.

**Table 1 pharmacy-08-00084-t001:** Characteristics of studies included in the review.

Study	Population	Region	Study	PrEP Available in Region at Time of Study?	Knowledge/Awareness	Comfort/Confidence	Intentions/Experience	Perceptions/Attitudes
Meyerson 2019 [13]	284/850 community pharmacies	Indiana USA	Survey	Y	56% reported familiarity with PrEP and 52% reported insufficient knowledge on the topic	54% were comfortable counseling PrEP patients	16% reported dispensing PrEP at their pharmacy and 12% had consulted about it	N
Okoro 2018 [14]	347/2815 community pharmacists	Minnesota USA	34-item survey	Y	52% were aware of FDA approval for PrEP. Most reported being at least somewhat knowledgeable about PrEP	Having some knowledge of PrEP was associated with being comfortable counseling PrEP	Reported never receiving inquiries about PrEP from clients (72%) and providers (81%)	56% disagreed that PrEP use would contribute to increased HIV transmission
Broekhuis 2018 [12]	140/1140 pharmacists	Nebraska, Iowa USA	18-item survey	Y	42% were familiar with the use of PrEP, but 25% were familiar with the CDC guidelines for its us	>80% were comfortable assessing HIV risk+ performing + counseling on HIV testing and providing counseling on PrEP use	12% had previously counseled on PrEP	Concerned about PrEP in terms of: time burden (61%), compensation (55%), skill set (39%), patient adherence (63%), loss to follow-up (56%) and promotion of drug resistance (51%)
Unni 2016 [17]	251/1392 community pharmacists	Utah USA	26-item survey	Y	>75% knew PrEP is indicated for those who have HIV-positive partners	20% agreed they had the ability + confidence to counsel	17% agreed/ strongly agreed when asked about the intention to counsel and a further	>90% agreed that counseling about PrEP is an ethical obligation and patients would benefit from counseling
Yoong 2016 [15]	59/160 pharmacists	Canada	27-item survey	N	94% were somewhat familiar or very familiar with the concept of PrEP	31% had insufficient familiarity of PrEP to provide counselling	42% had received a question about PrEP in the past year	Required PrEP to have a minimum efficacy of 70% to recommend
Sanchez-Rubio Ferrandez 2016 [11]	169 hospital pharmacists (no response rate)	Spain	31-item survey	N	46% of pharmacists had low or no familiarity with PrEP	N	N	Require PrEP to have a minimum efficacy of 86% before supporting its widespread use
Matyanga 2014 [18]	112/125 pharmacists	Zimbabwe	15-item survey	Y	58% were knowledgeable about PrEP	N	94% willing to stock + 87% intended to dispense PrEP if provided as a pharmacist-initiated medicine	77% disagreed that PrEP would lead to long term side effects
Shaeer 2014 [16]	225 pharmacists (no response rate)	Florida USA	30-item survey	Y	63% were unaware of CDC guidance on PrEP use and 59% were not aware of what the acronym PrEP represented	53% were comfortable counselling patients on PrEP	>80% had no inquiries from either patients or providers about PrEP. Only 22% had dispensed PrEP	68% perceived that PrEP encouraged high-risk behavior, 65% agreed PrEP would increase rates of other STIs and 26% reported PrEP would lead to increased rates of HIV

USA = United States of America, Y = yes, N = no.

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
