# Peer review of "A Narrative Review of Continuing Professional Development Needs for Pharmacists with Respect to Pre-exposure Prophylaxis (PrEP) for Human Immunodeficiency Virus (HIV)"

_pharmacy, 2020, doi:10.3390/pharmacy8020084_

Round 1

Reviewer 1 Report

This article reveals gaps in the knowledge of pharmacists in prevention of HIV by using the PrEP approach. Although interesting, some aspects deserve attention.

  1. As indicated in the paper, data evaluated is very short and most of them come from USA, so that it cannot be extrapolated to all pharmacist.
  2. In order to fulfill the gaps, a more detailed description of the knowledge and guidelines is required. Thus, the study will be improved giving details of the aspects of the Molecular Biology, posology, monitoring, patient support, etc.
  3. Why the authors did not ask the professional associations?
  4. Raw data accompanied with statistical analysis is necessary to support the conclusions, may be as a table or a graph.

Author Response

This article reveals gaps in the knowledge of pharmacists in prevention of HIV by using the PrEP approach. Although interesting, some aspects deserve attention.

  1. As indicated in the paper, data evaluated is very short and most of them come from USA, so that it cannot be extrapolated to all pharmacist.

Authors’ Response: Thank you for the comment, we have highlighted this in the paper as stated.

  1. In order to fulfill the gaps, a more detailed description of the knowledge and guidelines is required. Thus, the study will be improved giving details of the aspects of the Molecular Biology, posology, monitoring, patient support, etc.

Authors’ Response:  Thank you for this comment.  However, these topics are outside of the scope of this paper.  This paper is focused on CPD needs for pharmacists relating to PrEP.

  1. Why the authors did not ask the professional associations?

Author’s Response:  The authors are unclear of the meaning of this comment. This was meant to be a narrative literature review and therefore no professional associations were asked for data.

  1. Raw data accompanied with statistical analysis is necessary to support the conclusions, may be as a table or a graph.

Author’s Response:  Please see new figure to outline search strategy, as well as raw data within the category descriptions.

Reviewer 2 Report

Abstract

Line 16: An electronic search of databases and CPD-related journals was...

→ Please specify the databases that you searched for this review.

Introduction

Line 44: ... optimal care to patients seeking its use

→ optimal care to patients seeking its use.

Methods

  1. It would be more persuasive to conduct a review through following guidelines for systematic reviews by PRISMA's Transparent Reporting of Systematic Reviews and Meta-analyses found at http://www.prisma-statement.org/. Please use the PRISMA to write your methods and results sections.
  2. You have to provide a flow chart to describe your process of data collection.
  3. It was unclear your searching strategies for literature reviews. What were your inclusion and exclusion criteria for the literature review? Please provide more information to elaborate on this process.

Results

  1. You have to describe the process of literature searching. For example, how many studies did you found in the beginning? How many studies did you exclude per your exclusion criteria? Please report your results per PRISMA guidelines.
  2. Please describe your evaluation of the risk of bias for the literature review.

Table

Please indicate the study setting of each study listed in this table (e.g., hospital, community pharmacy).

Author Response

Abstract

Line 16: An electronic search of databases and CPD-related journals was...

→ Please specify the databases that you searched for this review.

Authors’ Response:  Added: PubMed, EMBASE, International Pharmaceutical Abstracts

Introduction

Line 44: ... optimal care to patients seeking its use

→ optimal care to patients seeking its use.

Authors’ Response:  Done.

Methods

  1. It would be more persuasive to conduct a review through following guidelines for systematic reviews by PRISMA's Transparent Reporting of Systematic Reviews and Meta-analyses found at http://www.prisma-statement.org/. Please use the PRISMA to write your methods and results sections.

Authors’ Response: This was not a systematic review and therefore PRISMA does not apply. We have, however, added a flowchart as suggested below.

  1. You have to provide a flow chart to describe your process of data collection.

Authors’ Response:  Done.

  1. It was unclear your searching strategies for literature reviews. What were your inclusion and exclusion criteria for the literature review? Please provide more information to elaborate on this process.

Authors’ Response:  No limits were placed on the search for study design, therefore literature reviews would have been identified. It is notable, however, that no reviews were found.

Results

  1. You have to describe the process of literature searching. For example, how many studies did you found in the beginning? How many studies did you exclude per your exclusion criteria? Please report your results per PRISMA guidelines.

Authors’ Response:  Please see Figure 1.

  1. Please describe your evaluation of the risk of bias for the literature review.

Authors’ Reponse:  This was a narrative review and not a systematic review. A systematic risk of bias assessment was therefore not conducted.

Table

Please indicate the study setting of each study listed in this table (e.g., hospital, community pharmacy).

Authors’ Response:  Done.

Round 2

Reviewer 1 Report

The authors have made some changes to clarify the aims and methodology, but my concerns remain:

  1. Data evaluated is very short. It should be increased, for instance consulting the professional associations.
  2. A more detailed description of the knowledge is required. Doing so we can have a clear idea of the needs.
  3. Raw data accompanied with statistical analysis is necessary in table 1.

Author Response

  1. Data evaluated is very short. It should be increased, for instance consulting the professional associations.

Authors’ response: As stated previously, this was not part of our pre-established protocol and therefore inappropriate to include. As well, it is unclear what professional associations the reviewer is referring to and what data we would obtain from them.

  1. A more detailed description of the knowledge is required. Doing so we can have a clear idea of the needs.

Authors’ response: Thank you. We have added more details to Section 3.1 – knowledge/awareness.

  1. Raw data accompanied with statistical analysis is necessary in table 1

Authors' response:  Added. 

Reviewer 2 Report

This is an interesting topic; the authors have addressed all review’s concerns in a clear way.

Title

Please indicate the narrative review in title.

Figure

Please check the accuracy of the number of “Total number of duplicate.” It may be 17 rather than 14.

Author Response

Title

Please indicate the narrative review in title.

Authors’ Response:  Done

Figure

Please check the accuracy of the number of “Total number of duplicate.” It may be 17 rather than 14.

Authors’ Response:  Changed.

Round 3

Reviewer 1 Report

Data evaluated is very short and this is a big limitation of the study. In addition to publications, consultation to pharmacists´s associations (hospital, community) about the needs could be a way to improve it.